# Evaluation of the Intergenerational Equity of Public Open Space in Old Communities: A Case Study of Caoyang New Village in Shanghai

**Zelin Zhang, Xiaomin Tang * and Yun Wang**

Department of Landscape Architecture, School of Design, Shanghai Jiao Tong University, Shanghai 200240, China; zzlsum@sjtu.edu.cn (Z.Z.); wangyun03@sjtu.edu.cn (Y.W.)
* Correspondence: xmtang@sjtu.edu.cn

**Abstract:** Public open space (POS) confers numerous physical and mental health benefits to people throughout life. The study applies POI and other multi-dimensional methods to examine the intergenerational equity of POS within a 15-min living circle of old communities in high-density cities. Firstly, an evaluation system for the comprehensive POS supply level in the community is constructed from the perspective of the quantity, quality, and spatial distribution of POS. Secondly, based on the idea of all-age sharing, the population in the community is divided into children, young and middle-aged, and elderly, and the factor "adaptation space share" is introduced to supplement the intergenerational equity evaluation system. The study takes Caoyang New Village in Shanghai as an example and the districts of the community committee as the basic evaluation units. The results show that the comprehensive supply of POS in Caoyang New Village is relatively high but there is still a mismatch between supply and demand; the intergenerational equity level is medium, and spatial alienation exists between POS supply and intergenerational equity; and the adaptation space share for children is much higher than that for the elderly, young and middle-aged people. Furthermore, young and middle-aged people are found to form a marginal group in spatial sharing and resource allocation. Finally, it is proposed that in community POS planning, attention should be focused on the differences in population age structure and on spatial cultural attributes and functional compounding on the basis of ensuring the comprehensive service of POS, so as to promote all-age sharing in community POS.

**Keywords:** intergenerational equity in public open space; all-age sharing; multidimensional big data; life circle; Caoyang New Village



## 1. Introduction

Public open space (POS) is open, public, and freely accessible and represents one of the basic types of land use that provides opportunities for people to engage in "conservation, recreation, and contact with nature" [1]. POS manifests in different forms such as green spaces, pocket parks, corner gardens, pedestrian streets, and gray spaces outside architectures for people to stay, etc. [2–6]. POS has widely been considered as an important contributor to promoting people's physical and mental health and emerged as one of the focal points of ecology [7,8]. As one type of public investment, POS is expected to serve communities in a fair way [9]. People should have access to these shared places regardless of their individual wealth and means, which is also one of the reasons why POS is usually highly valued [10]. This research focuses on those POSs with an area greater than or equal to 400 m$^2$ as the object.

In the context of rapid urban development, urban resources are not always evenly distributed [11]. There are many cases presenting strong evidence of the inequity in urban space, especially in old communities of high-density cities, which provide small living spaces [12]. In addition, there are many problems such as insufficient total POS, and

unbalanced allocation of facilities [13,14]. In this context, in the face of the people-oriented construction requirements of the new urbanization process, how to implement people-centered and equity principles in POS management has become an issue of wide concern.

There is a growing body of literature demonstrating that in the allocation of POS in urban communities, attention should be paid to the spatial needs and activity characteristics of different age groups [15]. Meanwhile, most current urban policies aim at restructuring the city to provide a better standard of living for all residents [16]. The World Health Organization (2021b) emphasizes the investment need for promoting intergenerational interactions [17]. In 2022, China's Ministry of Housing and Urban–Rural Development clearly proposed creating all-age friendly communities [18], also known as all-age communities, which refers to communities that, regardless of age, provide each of their residents with physical space such as housing, public services, outdoor environments, and shared social space; therefore, in such communities, residents can access opportunities for health, well-being, and public participation [19]. Based on this, the new idea and requirement for "all-age sharing" is proposed for the renewal of urban public space, emphasizing the enjoyment of equal opportunities and sharing of spatial services by residents of different age groups [20]. That is, all-age sharing aims to promote a fair intergenerational allocation of POS resources and further encourages positive social interactions across different generations [21].Therefore, this study applies the concept of "intergenerational equity" to characterize the all-age sharing of POS.

Within the study of social justice and the population aging trend, "Intergenerational equity", first proposed by the American scholar Page (1977), mainly concerns the welfare and resource allocation between the present and future generations [22], and with the development of research, "intergenerational equity" was introduced into the study of urban planning [23]. In terms of different perspectives on the time dimension, "intergenerational equity" can be divided into "equity between generations in the same time period" and "equity between generations in different time periods" [24] and is a core concept of sustainability, emphasizing the intergenerational integration of POSs, guaranteeing equal opportunities for all, common participation and sharing of results [25–27]. We contend the former in this review paper, i.e., it takes generations in the same time period as an object and studies their unbalanced spatial allocation and the interoperability of resources in POS shared by different generations. In this paper, human beings are divided according to age into three major generational groups, taking into account the relevant age classification of the United Nations Health Organization and the national situation of China, as follows: children (0 to 18 years old), young and middle-aged (19 to 59 years old), and elderly (greater than or equal to 60 years old)

## 2. Literature Review

A large number of scholars have studied POS in terms of its quantity, spatial distribution, vitality, accessibility, quality [28–30], etc., using methods such as overlay analysis, spatial awareness, expert knowledge, GIS network analysis, and the two-step floating catchment area method [31–33]. Such studies helped in discovering the importance of POS in enhancing the vitality and quality of cities [34]. In recent years, with the intervention of sociology in the study of POS, research has begun to focus on the "public attributes" of POS from a sociological perspective, taking into account the needs of residents in all aspects of the planning, design, construction, and governance of POS [35] and studying POS in its four dimensions: public participation, publicness, inclusiveness, and equity [36]. In the planning of POS, more emphasis is being placed on resident participation in spatial governance [37] with a focus on spatial inclusiveness [38], such as whether the space is child- and age-friendly and has spatial equity [39–41].

Research on the equity of POS has gone through four main periods: territorial equality, spatial equity, social equity, and social justice [42,43]. First, territorial equality explores the equity of spatial quantity and scale from a supply perspective, emphasizing equal distribution as a core objective and equal access for all [44,45], which is usually measured

by per capita indicators [46]. At this stage, territorial equality does not take into account the characteristics of human needs, the physical spatial layout of facilities, and the effectiveness of the service. Then, from the perspective of supply and demand balance, accessibility has been introduced in studies of spatial equity to emphasize the utilization efficiency of POS [47], while further attention has been paid to the level of space quality and service [48,49]. Commonly used measures of research on spatial equity include the Mann–Whitney U test, ordered correlation analysis, analysis of variance, bivariate spatial autocorrelation, and regression analysis [50,51]. Accessibility calculation methods of POS in research on spatial equity usually include the minimum distance method, the buffer distance method, the gravitational model method [52,53], etc. In addition, evaluation systems and audit tools are normally used to assess environmental quality, including the satisfaction questionnaire [54], Public Open Space Tool (POST) [55], Public Life (PSPL) tool [56], etc. In this stage, spatial equity measurement has been greatly improved in terms of methods, and the focus has shifted from the larger scale of administrative districts to the smaller scale of neighborhoods, making the equity research more detailed and accurate [57]. However, research focusing on spatial equity ignores the differences in social space and user groups. Additionally, the perspective of social equity focuses on the differences brought about by disparities in socioeconomic characteristics, such as settlement differences, income differences [58], and differences related to specific populations [59], and it mostly evaluates the social equity performance of public spatial distribution by the Lorenz curve and Gini coefficient in economics [60]. However, these usually focus only on the elderly or children, and few studies have compared spatial needs based on all ages, i.e., spatial intergenerational equity [61]. The last perspective of social justice focuses on socially disadvantaged people and emphasizes spatial resources for specific disadvantaged groups [62–64]. Unlike the perspectives of equity mentioned above, spatial justice emphasizes the consequential spatiality of social justice as well as incorporates the considerations of democracy and human rights [65]. The general trend of research has shifted from equity in land to equity for people [66]. The overall research ranges from provincial and urban scales to community scales, but few studies have focused on spatial equity at the 15-min living circle scale.

To help in filling some of the knowledge gaps mentioned above, this research is, based on the concept of equity and discussions of the evaluation and analysis methods of intergenerational equity. Our research aims to develop an evaluation method to assess the equity among different age groups within the 15-min living circle of old communities in high-density cities and to further discuss and elaborate on the entanglement between urban regeneration and people's different needs for community POSs in recent years. In this study, the first village for workers in China, Caoyang New Village in Shanghai, was selected as the community for evaluating the intergenerational equity of POSs within the 15-min circle, providing a reference for future policy formulation and an evaluation model that is applicable to other cities.

The balance between supply and demand for POS is an important part of equity theory. In this paper, we first consider residents as undifferentiated individuals and analyze the supply and demand characteristics. In terms of POS supply, based on the above research review, the supply factor of POS usually includes space area, quality, quantity and facilities and equipment, and these factors would directly affect the experience and enjoyment of residents in POS. We find that studies often use spatial quantity or layout characteristics of POS to measure its spatial supply, while some scholars consider the difference in spatial supply in terms of quality [48], but in general, it is still difficult to comprehensively characterize the comprehensive grade level of space supply. Hence, we select three indicators to comprehensively characterize the level of POS supply in the community—quantity, spatial distribution, and quality—and draw on the methods used for measuring these indicators in previous studies. The three indicators are progressive: firstly, from the supply point of view, the space supply is measured by using the "Recreation opportunity index", which reflects the evenness of the POS quantity supply, and the measurement method based on accessibility can reflect the actual quantity of space obtained

by residents more truly [67]; secondly, based on the relationship between supply and demand, taking into account the differences in the spatial distribution of residents, the method of the "location entropy" of the district is used to reflect whether the spatial layout is reasonable [57]; finally based on the concept of people-oriented urban development, the spatial quality supply is further measured on the basis of quantity and layout, reflecting the level of spatial services obtained by residents [68]. Here, we choose POST (Public Open Space Tool), a standard audit tool, to objectively measure the quality of POS [55], and its evaluation capability has been confirmed by several studies. Meanwhile, we introduce a Gaussian distance decay function, indicating that the POS quality service level decreases with increasing distance [48].

However, even the high level of comprehensive POS supply does not represent equal resource allocation for different age groups, and the generalized resource allocation model sometimes ignores the inequity of spatial resources due to the diverse needs of different age groups [69,70]. Therefore, from the perspective of all-age sharing and based on equity-related theory [71], the concept of "adaptation space share" is introduced to construct an intergenerational equity evaluation system. "Adaptation space share" means whether the POS provided meets the usage requirements and preferences of residents. Whether the POS is suitable or not requires consideration of various factors, such as the configuration of facilities [39], climatic conditions, accessibility [72], residents' satisfaction [73] and physical health condition [74], etc. Here, we focus on the difference in demand for space facilities and measure the area of POS, which can provide suitable facilities for different age groups. We consider the space needs of children, young and middle-aged people, and the elderly and conduct a comparative study on the supply of space to be adapted for these three age groups. Whether the supply of the adaptation space share is balanced is used as a criterion for intergenerational equity. As few studies have conducted all-age equity comparisons, the coefficient of variation in statistics is creatively drawn upon to characterize the degree of difference in the supply of adaptation space for the three groups. The coefficient of variation can eliminate the effects of scale and magnitude, and can objectively reflect the degree of difference between the three age groups in terms of the adaptation space share. The greater the variation is, the lower the intergenerational equity, and vice versa (Figure 1).

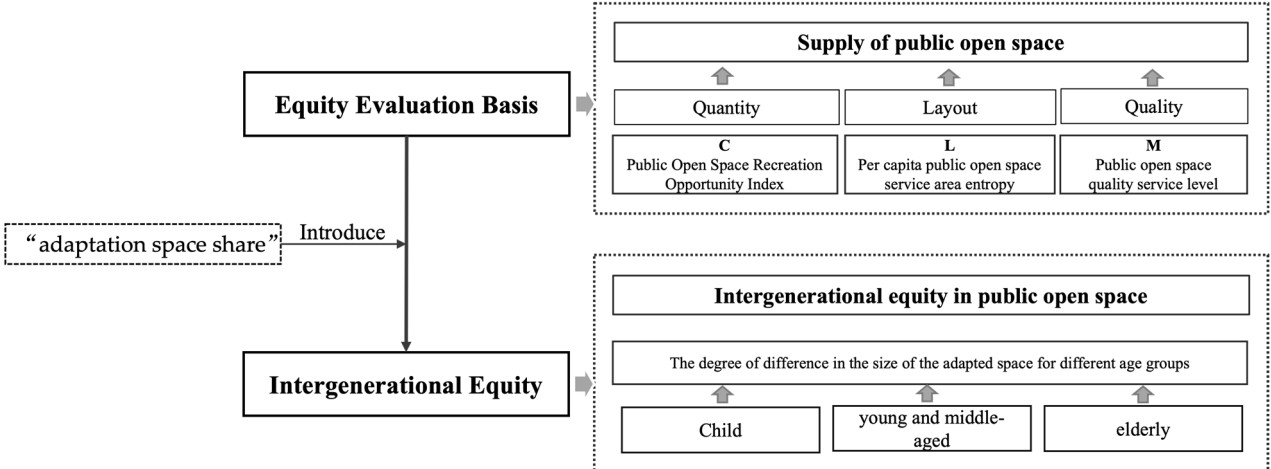

**Figure 1.** Methodological system for evaluating intergenerational equity in POS.

## 3. Methods

### 3.1. Study Area

Caoyang New Village, a street district in Putuo District, Shanghai, was built in 1951 and is the first new village for workers in China and the first large residential area established according to a "neighborhood unit". Its development process and spatial form can represent the general urban residential area in China [75]. It covers an area of 2.14 km² and has 20 neighborhood units under its jurisdictions (Figure 2). The resident population is

89,300 people, and the population density reaches 41,700 people/km$^2$. It is a typical old community in a high-density city [57].

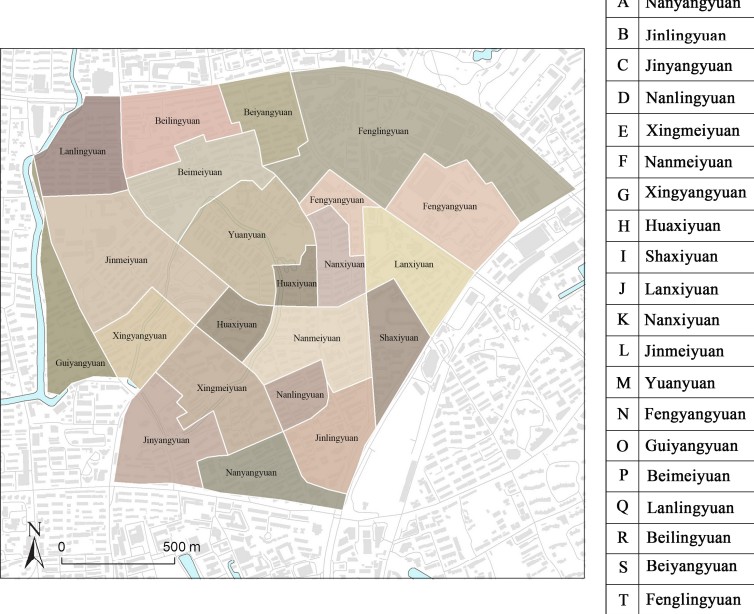

| | |
|---|---|
| A | Nanyangyuan |
| B | Jinlingyuan |
| C | Jinyangyuan |
| D | Nanlingyuan |
| E | Xingmeiyuan |
| F | Nanmeiyuan |
| G | Xingyangyuan |
| H | Huaxiyuan |
| I | Shaxiyuan |
| J | Lanxiyuan |
| K | Nanxiyuan |
| L | Jinmeiyuan |
| M | Yuanyuan |
| N | Fengyangyuan |
| O | Guiyangyuan |
| P | Beimeiyuan |
| Q | Lanlingyuan |
| R | Beilingyuan |
| S | Beiyangyuan |
| T | Fenglingyuan |

**Figure 2.** Names and distribution of units under jurisdiction of the residents. (In the later figure, the names of the unit will be represented by pairs of English letters).

As a typical high-density urban old community, Caoyang New Village has common problems of old communities in general, such as insufficient supporting facilities, inadequate spatial accessibility, uneven and unfair spatial distribution, etc. The POS of Caoyang New Village unfolds with the Huanbang green belt as the core. Due to the closed roads in some group districts, the travel environment is not sufficiently ordered and the POS connection between different districts is weak, resulting in low utilization of POS. Studies have been conducted to evaluate the comfort of walking space and POS renewal performance in Caoyang New Village [76,77], but few studies have been conducted to evaluate the POS in Caoyang New Village from the perspective of equity. Therefore, under the perspective of all-age sharing, we try to evaluate the POS of Caoyang New Village from the perspective of equity, in order to summarize the typical problems in old communities in high-density cities and provide a useful reference for subsequent renewal.

Caoyang New Village is one of the first "15-min community living circle" pilot communities in Shanghai. To address the two shortcomings of unsatisfied spatial quality and inadequate spatial governance, the government focuses on planning spatial coordination and resource policy supply and makes full use of spatial information technology to improve the level and service functions of facilities, such as elderly care, sports, and leisure. In particular, the renovation and activation of POS have become a top priority [78]. In China, such types of renewals are popular in old communities and urban villages in some urban central areas that face complex demographic structures, rigid social relationships, limited space resources, and prominent aging problems [79]. Therefore, evaluation of the equity of POS in Caoyang New Village has a certain practical significance.

*3.2. Data Collection*

3.2.1. Population Distribution Data

In order to accurately characterize the demand distribution of residents on the street scale, we obtained detailed data of the resident population in the seventh national census by visiting and researching the Caoyang New Village street office: the total population of Caoyang New Village is 89,302, including 8067 children, 49,534 young and middle-aged

people, and 31,701 elderly residents. Treating the residents as undifferentiated individuals, we found that the higher demand units are mostly concentrated in the peripheral areas. The distribution of demand is similar for different age groups. The distribution of the population in different jurisdictions and the distribution of the population by age groups are shown in Figures 3 and 4.

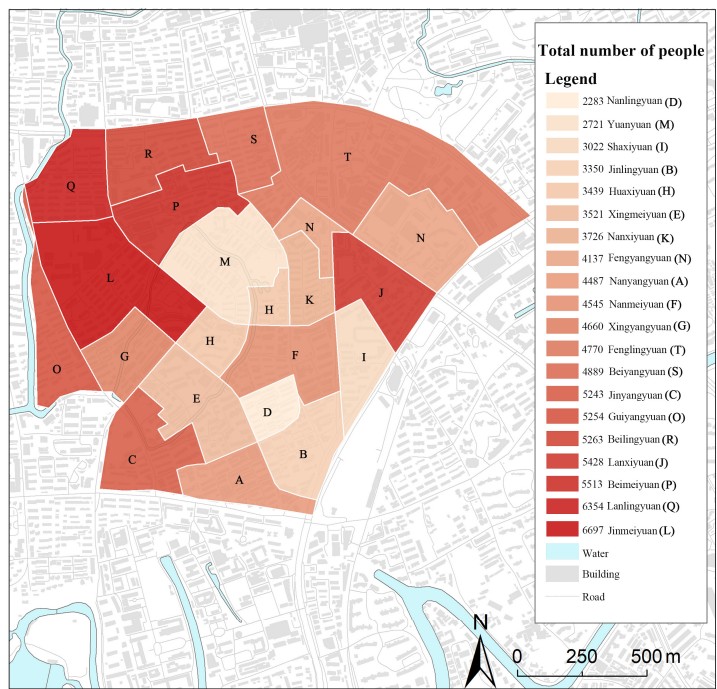

**Figure 3.** Population distribution of the units under jurisdiction of the residents.

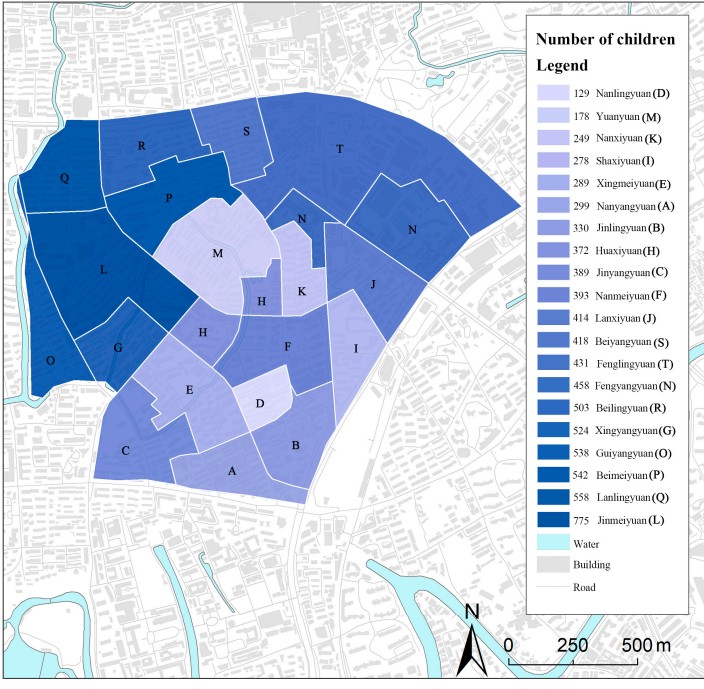

(**a**)

**Figure 4.** *Cont*.

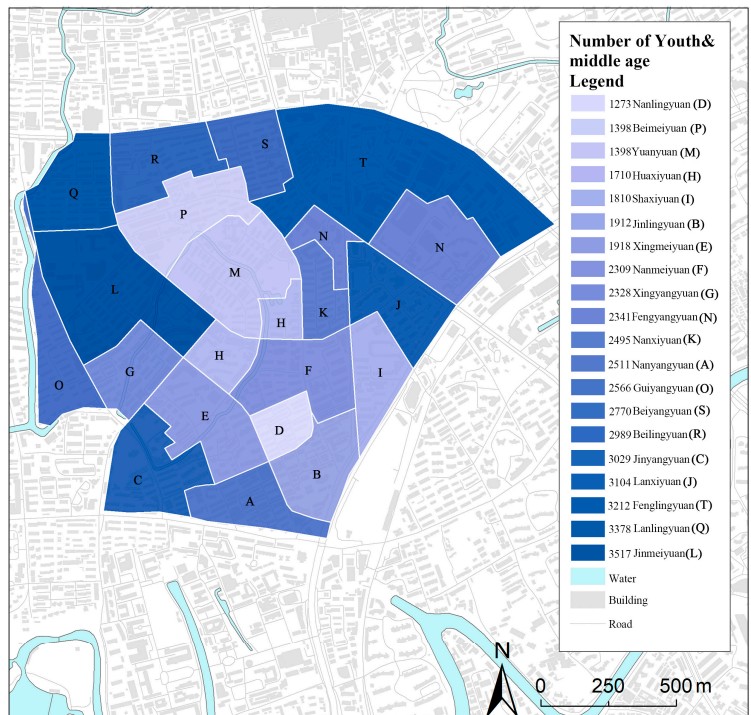

(**b**)

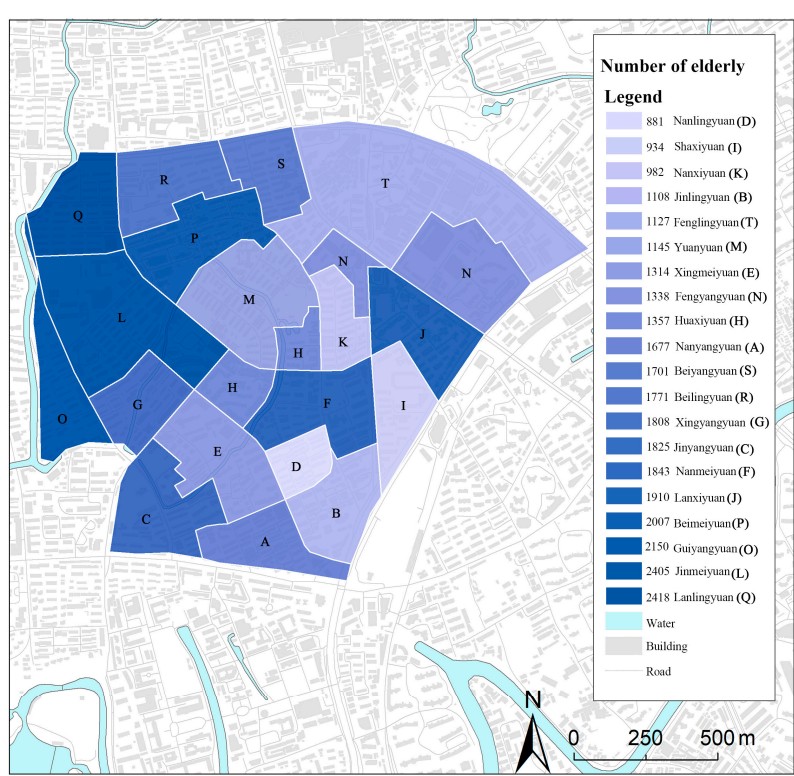

(**c**)

**Figure 4.** Number of (**a**) children, (**b**) young and middle-aged, and (**c**) elderly residents.

### 3.2.2. Delineated 15-Min Living Circle Range

The 15-min community living circle is a basic element of community living proposed by Shanghai in 2016, which represents "a safe, friendly and comfortable platform for basic social living within 15 min' walking distance, with basic services and POS for living" [80]. In order to break the pattern of dividing the "original community" by the scope of the living circle and the ambiguity of the scope caused by the inaccuracy of the "starting point" of the living circle [81], Python was used to crawl the POIs (points of interest) of the entrances and exits of residential neighborhoods in Baidu Map as the starting point of the living circle, and 283 entrances and exits of residential neighborhoods were obtained. Next, the data that are obviously not the entrances and exits of neighborhoods (such as school gates) were filtered out, and 179 entrances and exits of residential neighborhoods were finally identified. Combined with the lightweight route planning service (Direction Lite API, DLAPI) in Mapbox, the 15-min walking isochrone circles in the Isochrone API (https://docs.mapbox.com/playground/isochrone/, accessed on 23 February 2022) in Mapbox were obtained in batches, and in this isochrone circle, the walking calculation method of such isochrone circles was selected as the fastest path applicable to human walking, taking into account the road network and its quality, and feasibility, and the influence of the terrain on walking, which can more truly reflect the actual living range of residents. A total of 179 15-min walking isochrones were obtained by crawling. At last, ArcGIS 10.5 (Geographic Information System 10.5) was used to merge the isochrones to obtain the 15-min living area of the street (Figure 5).

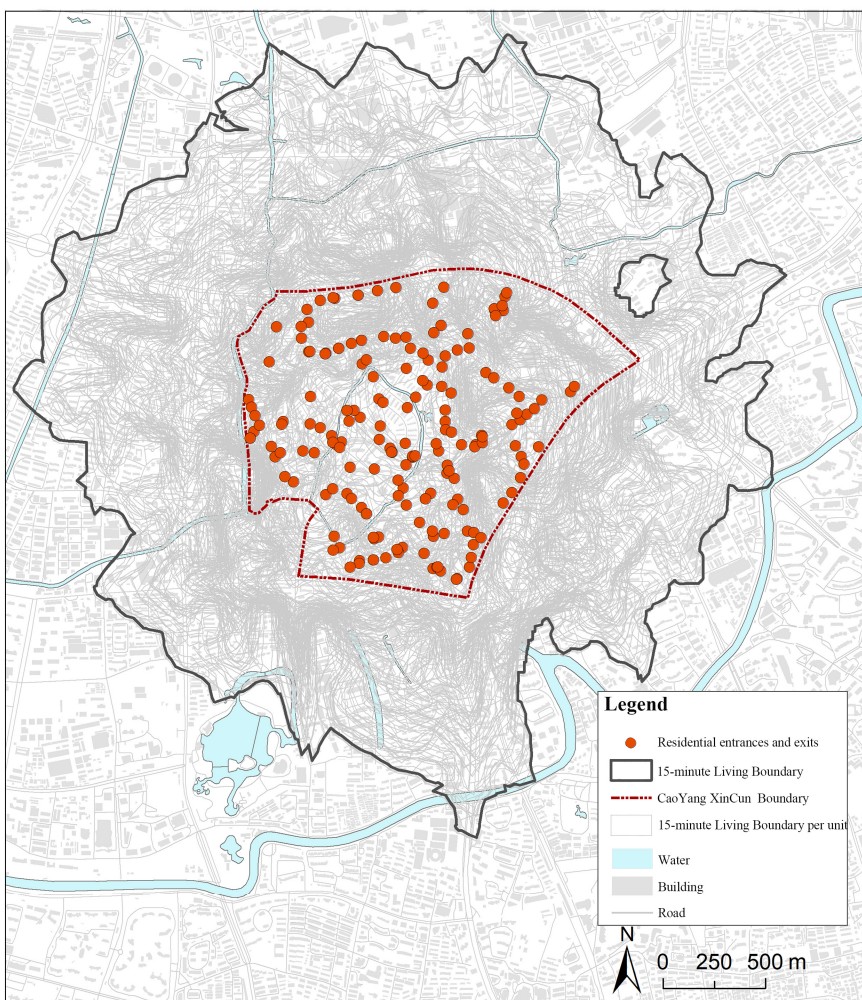

**Figure 5.** Caoyang New Village 15-min living circle area.

### 3.2.3. POS Data

Originally, all POSs inside the district boundary of the 15-min living circle were selected. POS data were crawled using Python and checked through field research. Finally, 28 public spaces larger than 400 m$^2$ were screened out (Figure 6). For the accurate measurement of POS quality, we subdivided them into three types according to the differences in four aspects—activity supply, environmental quality, convenience facilities, and spatial security—and combined these with the scale of POS. According to the above aspects, we concluded there was 1 POS of type I, 9 POSs of type II, and 18 POSs of type III (Table 1).

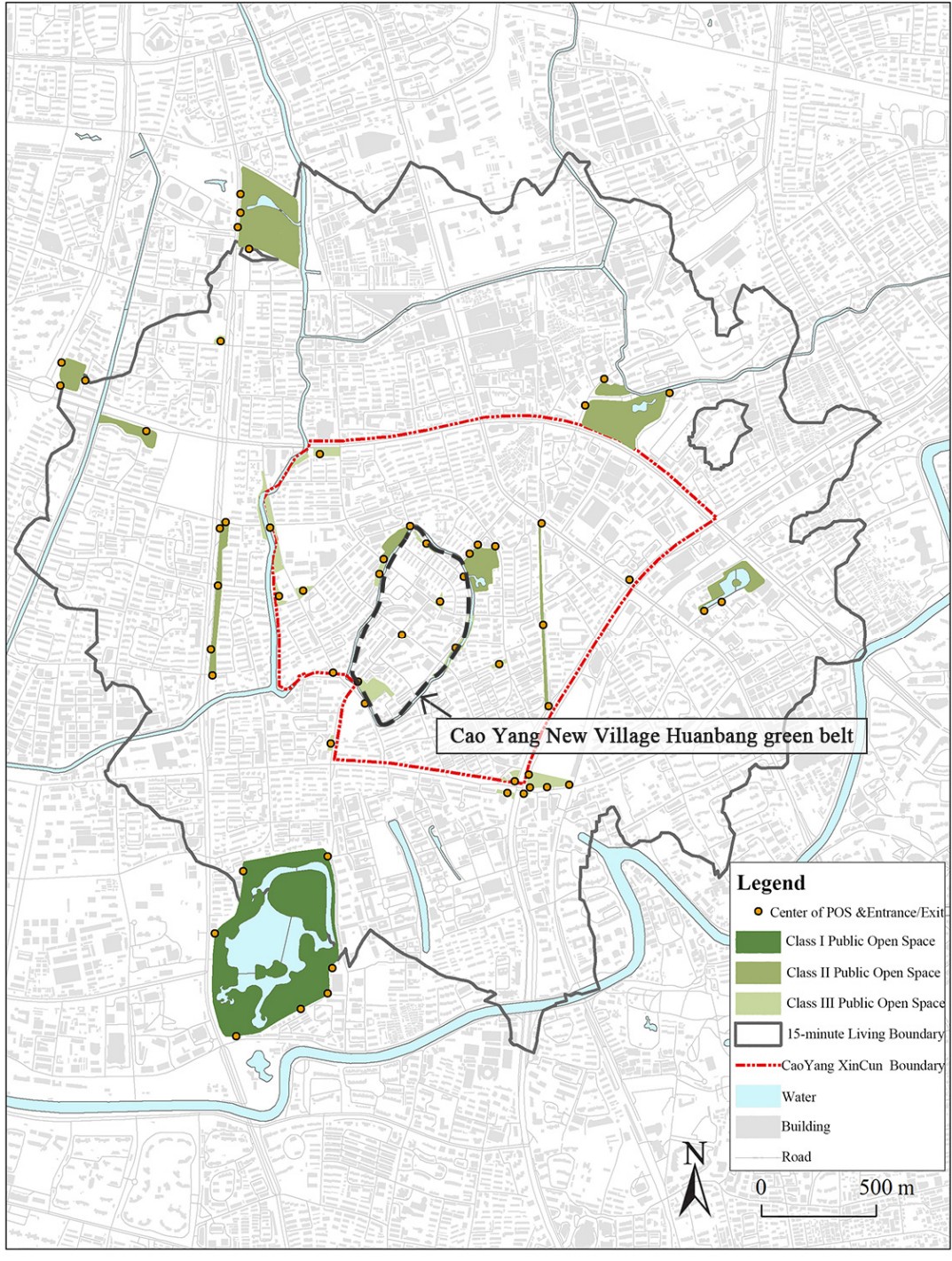

**Figure 6.** Distribution of 3 types of POS in Caoyang New Village 15-min living circle area.

**Table 1.** List of public open space categories and information within the 15-min living circle of Caoyang New Village.

| Types of POS | Definition | Name of POS |
|---|---|---|
| Type I | Refers to a large public open space greater than 5 hm$^2$ that can provide urban residents or visitors with functional facilities required for various recreational activities such as sightseeing, fitness, games for children, sports and science popularization, etc., mainly as comprehensive parks and green areas attached to large public buildings. | Changfeng Park |
| Type II | Refers to a medium-sized public open space of 1~5 hm$^2$ in the city with independent land use, basic recreation and service facilities, mainly for the residents in the 15 min living circle to carry out daily leisure activities nearby, such as community parks and green areas attached to small and medium-sized public buildings. | Wuning Park<br>Zhenru Park<br>Caoyang Park<br>Baixi Park<br>Begonia Garden<br>Daduhe Road Street Green<br>Shanghai West Workers' Cultural Palace<br>Lanxi Youth Park<br>Meichuan Park |
| Type III | Refers to a small public open space between 400 and 10,000 m$^2$, oriented to human needs, which occupies the unused space in various types of urban land to meet human needs, and provides people with leisure and recreation, sports and fitness, science education, and other services of limited open space, mainly as pocket parks, the green space attached to small public buildings. | Intersection of Yangliuqing Road and Meiling North Road<br>Zaoyang Park<br>Front Square of Putuo District Government<br>Intersection of Lanxi Road and Wuning Road<br>XingShan Green<br>Gui Xiang Green<br>Intersection of Meichuan Road and Yangliuqing Road<br>Huaxi Green<br>Intersection of Caoyang Road and North Zhongshan Road<br>Intersection of Meichuan Road and Xiqiujiang River<br>Yuan park<br>Front Square of Caoyang Book City<br>Intersection of LiCun Road and Meiling South Road<br>Intersection of Jinshajiang Road and Zaoyang Road<br>Intersection of Zhongshan North Road and Jinshajiang Road1<br>Intersection of Zhongshan North Road and Jinshajiang Road2<br>Intersection of Zhongshan North Road and Jinshajiang Road3<br>Shanghai Global Port Front Square |

3.2.4. Accessibility-Based Measurement of POS Service Coverage Area

This study used the ArcGIS 10.5 platform, setting the POS entrances and exits as the starting point (for small POS without obvious entrances and exits, the center of mass was regarded as the starting point). A network analysis method was applied: red light resistance was set as 30S in road intersections and 1 m/s walking speed was selected to construct network data sets. Finally, this study managed to obtain the coverage of POS services (Figure 7).

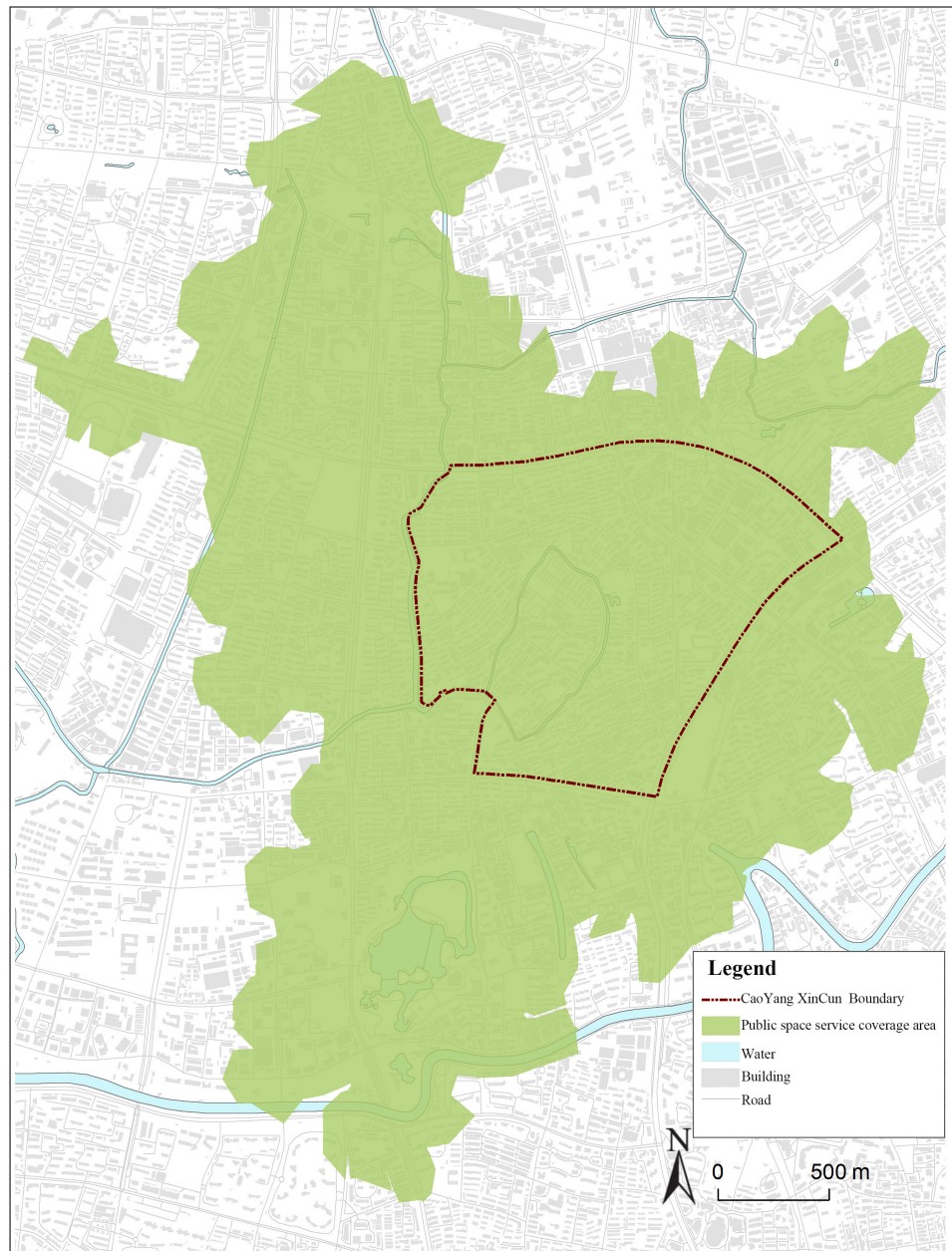

**Figure 7.** POS service coverage.

*3.3. Measurement*

3.3.1. Measurement of Public Open Space Supply Level

1.    Recreation opportunity index (*C*).

The recreation opportunity index (*C*), which reflects the degree of opportunities for recreation that residents can choose, where the higher the number of POSs under the same POS service coverage, the higher the POS recreation opportunity index (*C*), was calculated as follows:

$$C = \frac{A_{re}}{S} \times 100\% \tag{1}$$

where $A_{re}$ is the total service coverage area of the POS of the unit based on accessibility, and the overlapping part of the service coverage area is overlay calculated; and *S* is the area of the unit under the jurisdiction of the neighborhood committee.

2.    POS service location entropy (*L*).

POS service location entropy (*L*) is a unit measuring the POS distribution based on supply and demand and is calculated as follows:

$$L = \frac{A_{re}/P_d}{A_r/P_s} \tag{2}$$

where $A_r$ is the total service coverage area of POS within the Caoyang New Village based on accessibility, the the overlapping part of the service coverage area is overlay calculated; $P_d$ is the number of the actual population in the unit under the jurisdiction of the committee; and $P_s$ is the number of the actual population in the study area. When the value of $L$ is greater than 1, it means that the per capita level of POS enjoyed by the unit is higher than the overall average level; when $L < 1$, it means that the per capita level of POS enjoyed by the unit is lower than the overall average level.

3.   POS service quality level ($M_t$).

POST (Public Open Space Tool) was used to measure the quality of POS. Several studies have confirmed POST's ability and reliability to evaluate spatial quality comprehensively [82,83] using four aspects: activity supply, environmental quality, convenient facilities, and spatial safety [1]. Yet, the existing POST cannot be applied to the systematic assessment of the quality of community POS with diverse types and characteristics and widely varying functional and facility configuration requirements. According to Chinese POS standards and classification criteria, we removed several POST items that were irrelevant to the POS of Chinese cities (for example, the presence of barbecues and picnic tables) and optimize POST according to the size and functional characteristics of the POS in the study area, and weights were assigned to the measurement index using the expert scoring method, on the basis of which an integrated quality measurement model (IQM) of POS quality is constructed.

$$q_i = \sum_j A_j \times W_j \tag{3}$$

Here, $A_j$ denotes the score of the *j*th indicator, $W_j$ denotes the weight of the *j*th indicator of the node, and the scores of the four dimensions were superimposed to obtain the total score "$q_i$" of the quality of a certain POS node. Three types of spatial quality scores were measured (Figures 8 and 9).

The unit POS service quality level ($M_t$), which reflects the average POS quality service level within the 15-min living circle, was calculated by adding the Gaussian distance decay function $G(dit)$ to reflect the change in the POS quality service level with the increase in distance, considering that there is a certain difference in distance between the POS and the unit. It was calculated as follows:

$$G(dit) = \frac{e^{-\frac{1}{2} \times \left(\frac{d_{ij}}{d_0}\right)^2} - e^{-\frac{1}{2}}}{1 - e^{-\frac{1}{2}}} \tag{4}$$

$$M_t = \frac{\sum_{t \in d_{it} < d_0}^{N_t} [q_i G(d_{it})]}{N_t} \tag{5}$$

where $M_t$ is the average quality service level of the unit POS, *i* denotes the POS, $N_t$ denotes the number of POSs available to unit *t* in the limit travel time, and qi is the quality score of POS *i*. The Gaussian distance decay function was chosen for time cost, $d_0$ is the distance traveled in the limit time of 15 min of travel, and the walking speed is 1 m/s.

To make the data more comparable, the index results were divided, using the natural breakpoint method, into the 5 levels of high, relatively high, medium, relatively low, and low, with scores assigned from a high of 5 to a low of 1 and obtained with *C'*, *L'*, and *M'* (Figure 10).

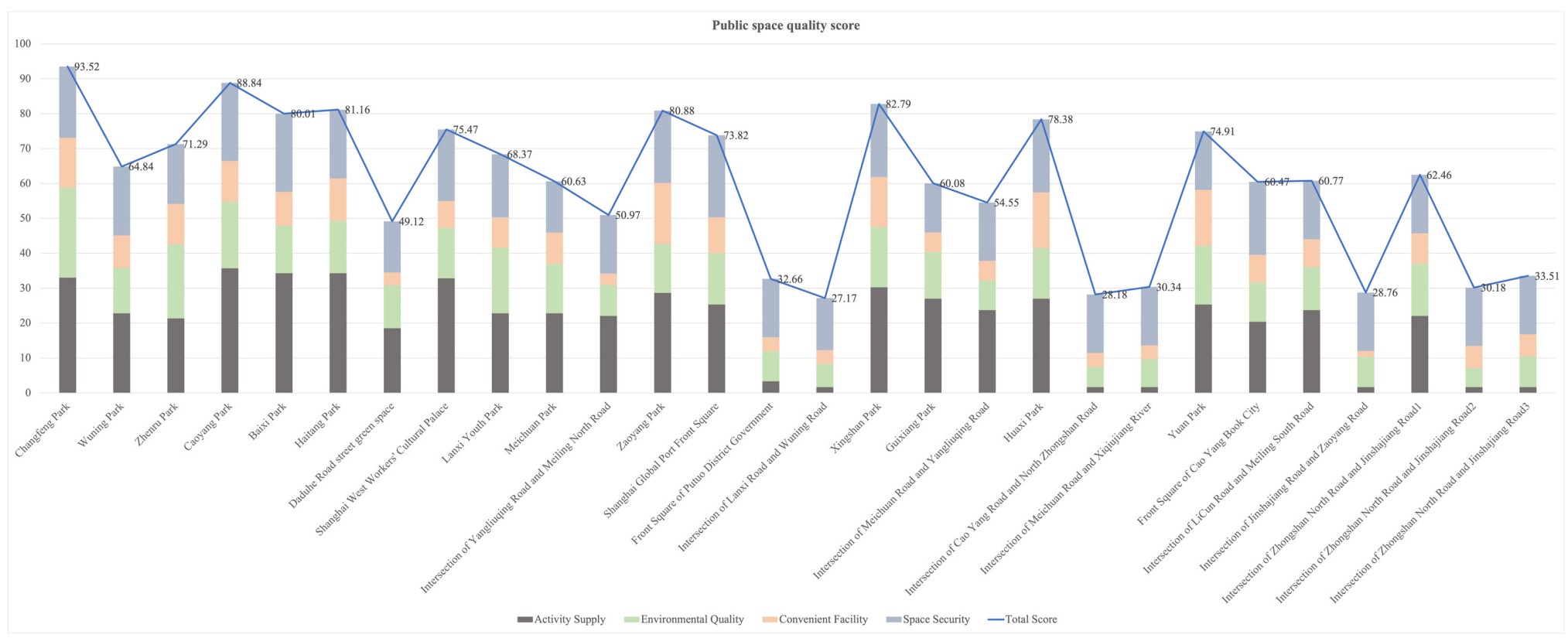

**Figure 8.** Public open space quality score.

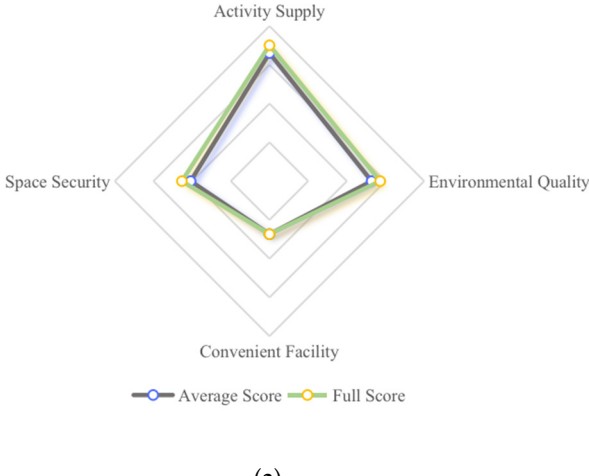

(**a**)

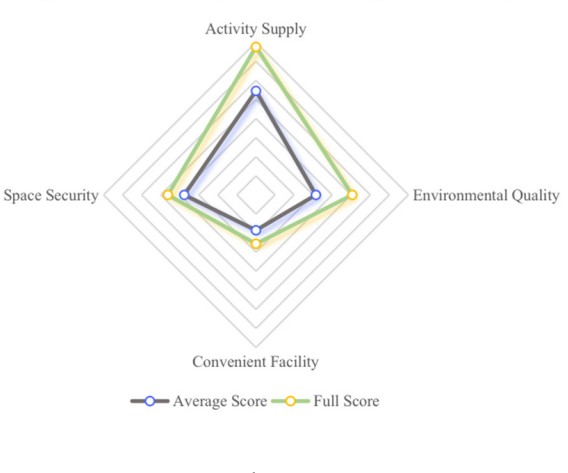

(**b**)

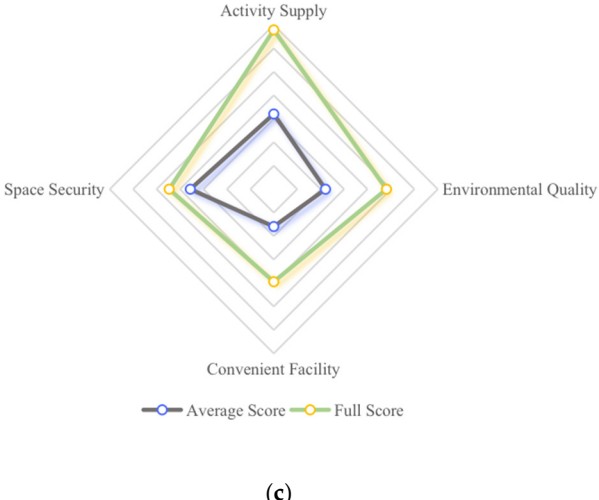

(**c**)

**Figure 9.** The average score of scoring items in four dimensions for space quality of the following different types of POS: (**a**) type I; (**b**) type II; (**c**) type III.

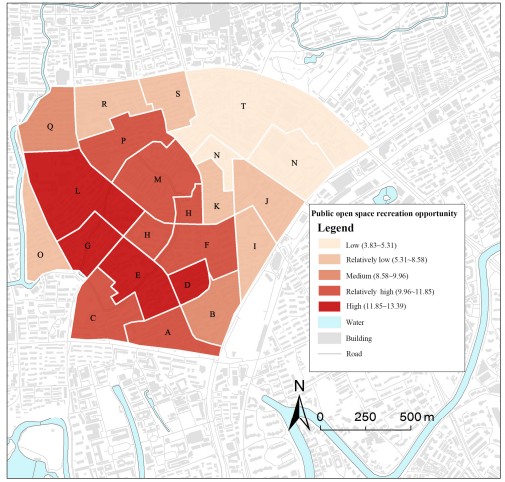

(**a**)

(**b**)

(**c**)

**Figure 10.** Quantity, layout, and quality of public space in the units under the jurisdiction of the residents' committee: (**a**) recreation opportunity index; (**b**) public space service area entropy; (**c**) public space quality service level. (Letters indicate different units of Caoyang New Village).

4. Comprehensive POS supply level (POS supply).

The comprehensive spatial supply is determined by the recreation opportunity index (*C*), POS service location entropy (*L*), and POS quality service level (*M*) together (Figure 10), and the variance inflation factor (VIF) of each index in Eq. is less than 5. According to the covariance judgment standard [84], the covariance problem is negligible when VIF < 10, so it is obtained by the weighted average of each index using the value range [0, 5]. The calculation formula is as follows:

$$S(pos) = \left(C' + L' + M'\right)/3 \qquad (6)$$

3.3.2. Evaluation of Intergenerational Equity

1. Criteria for judging the adaptation of POS.

Based on the activity facilities preferences of people in different age groups and the corresponding spatial elements, the area of activity space in POS that is suitable for that age group is counted and referred to as the adapted area of POS. The study summarizes the research of the preferences activities of different age groups in China through the literature research method used as the basis for judging the suitable space for different age groups. According to related research, we summarized that children tend to prefer play facilities such as ball games, skateboards, sandboxes and wooden horses, and that diverse environments such as activity lawns and water bodies are more attractive to children [85–88]. Reffering to the young and middle-aged groups, these two groups have a similarity in space type use preference, usually using fitness equipment, jogging track, etc. for fitness and exercise activities, followed by activities such as resting and viewing [89,90]. Elderly people prefer spaces that can provide activities such as chess, talking, resting, square dancing, enjoying the scenery, walking, equipment fitness, watching children, social dancing, etc. [91,92].

Based on the research above, our research divided the activity space into 3 major categories: exercise and fitness, leisure and entertainment, and culture and education. According to the crowd activity preference, there are 10 types of adapted spaces for the elderly, 12 types of adapted spaces for young and middle-aged people, and 11 types of adapted spaces for children (Table 2).

**Table 2.** Classification of activity sites in public spaces and criteria for judging the adaptation of activities.

| Activity Space | | Suitable for People | | |
| --- | --- | --- | --- | --- |
| **Classification** | **Specific Space Type** | **Elderly (≥60 Years Old)** | **Middle-Aged and Youth (19–59 Years Old)** | **Children (0–18 Years Old)** |
| Types of exercise and fitness | Health trails/walking paths | ● | ● | ● |
| | equipment fitness space (double bar, Tai Chi hand pusher, etc.) | ● | ● | / |
| | Multifunctional sports space (can be used for two activities or more at the same time, interactive game) | ● | ● | ● |
| | Ball game space | / | ● | ● |
| | Roller skating/skateboard space | / | ● | ● |
| | Rock climbing activity space | / | ● | ● |
| | Activity lawn | ● | ● | ● |
| | Children's play space (sand trap) | / | / | ● |
| Types of leisure and entertainment | Casual chess space | ● | / | / |
| | Quiet resting space (space for pavilions, resting seats, etc.) | ● | ● | ● |
| | Meeting and networking space (square dancing/socializing) | ● | ● | / |
| | Horticultural growing space | ● | ● | ● |
| Types of cultural education | Social education venues (legal publicity, etc.) | ● | ● | ● |
| | Nature education venues (reading rooms, etc.) | ● | ● | ● |

Description: "●" means suitable for this age group, "/" means not suitable for this age group.

2. Adaptation space share ($Q$).

The adaptation space share ($Q$), i.e., the allocation of adaptation space resources of Caoyang New Village resident district unit for different age groups within the 15-min living circle, reflects the friendliness level of the POS toward different age groups. Considering that a single POS will serve multiple units at the same time, the space-sharing situation is taken into account when calculating the unit's adaptation space share, and the calculation formula is as follows:

$$Q = \sum_{i=1}^{n} \frac{S_r}{P_n} \tag{7}$$

where $S_r$ is the area of the POS for a certain age group, and $P_n$ is the total number of people in this age group in the common unit of POS.

3. Intergenerational equity ($F$).

Intergenerational equity ($F$) reflects the equity level of the POS distribution under the differentiated needs of different age groups in the 15-min living circle. The lower the variability in access to the adaptation space of the POS for different age groups, the higher the intergenerational equity is, and vice versa. The degree of variability of these three age groups in terms of adapted space is expressed by the coefficient of variation ($CV$), which is calculated as follows.

$$CV = \frac{\sigma}{\mu} \tag{8}$$

$$F = \frac{1}{CV} \tag{9}$$

$\sigma$ is the standard deviation, and $\mu$ is the unit of different age stage space to obtain the average. The natural breakpoint method was used to divide intergenerational equity into the 5 levels of high, relatively high, medium, relatively low, and low, where the score from high to low is assigned as 5~1 to obtain $F'$.

## 4. Results

### 4.1. The Average Supply Level of POS in Caoyang New Village Is Relatively High, but There Is Still a Mismatch between Supply and Demand

The average level of comprehensive spatial supply (mean score 3.05) in Caoyang New Village is relatively high (Figure 11). There are 45% of units concentrated in the northeast area that fall short of the average level, and 55% of the units concentrated in the southwest that are above the average level. There is a significant bifurcation in the geographical distribution of POS supply, and there is an obvious problem of mismatch between supply and demand, with a high supply area dominated by the green belt around the waterfront, but its demand is low. This is mainly because the planning model of the "neighborhood unit" in Caoyang New Village makes it difficult for the existing POS layout and scale to break through the original urban spatial structure. Accordingly, most of the POS is concentrated in the Huanbang green belt area, while for the rest of the urban area it is difficult to achieve increased space to provide more POS due to the high-density building layout.

On the other hand, the results of POST show that the average quality score of 28 POSs within the 15-min living circle of Caoyang New Village is only 60.15. The average quality score of different types of POS can be ranked as follows: type I space > type II space > type III space. The highest quality score is 94.52 points for a type I space, Changfeng Park (Figure 12a). The lowest quality score is for a type III space, Shanghai Global Port Front Square (Figure 12b), with a score of only 27.17. From the four dimensions of POST, there is still great room for improvement in the diversity of activity space and the quality of the environment of Caoyang New Village.

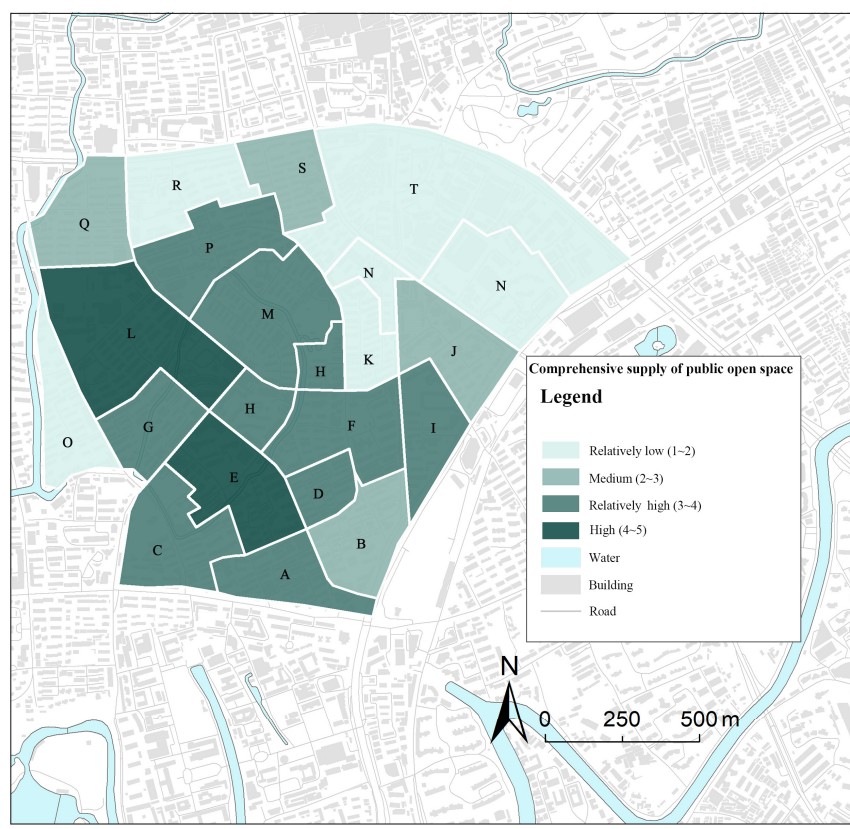

**Figure 11.** Comprehensive public open space supply (Letters indicate different units of Caoyang New Village).

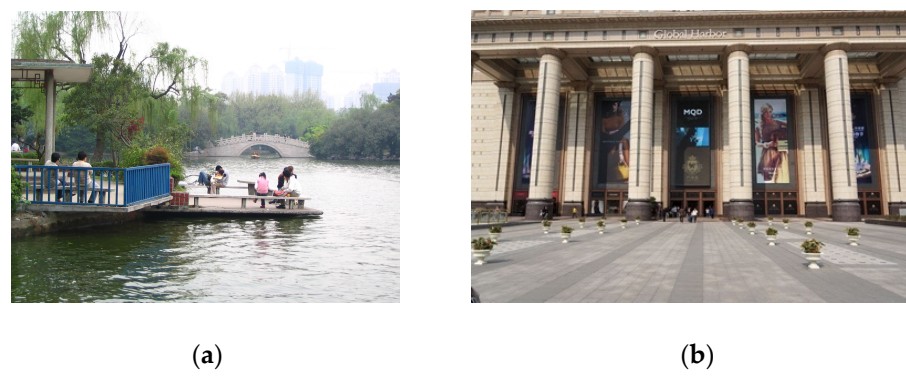

**Figure 12.** (**a**) Changfeng Park and (**b**) Shanghai Global Port Front Square.

*4.2. Average Intergenerational Equity Is Medium, and over Half of Units Have below-Average Intergenerational Equity*

The average level of intergenerational equity (mean score of 2.8) of Caoyang New Village is medium (Figure 13), and 55% of the units are below the average, showing a decreasing distribution trend from northwest to southeast. On the whole, there are two main reasons for the differences in intergenerational equity: ① Incompatible intergenerational use of POS, manifested as one activity space with clear functions usually serving for a single age group, and insufficient considerations for intergenerational sharing of composite spaces. For example, although type I spaces have a clear functional zone and a variety of activity spaces, children's needs here are usually ignored in such composite spaces, and there is a large gap in terms of adaptation space shared among children and the two other age groups, such as in the case of Changfeng Park; type II spaces generally serve a single age group and lack buffer space between different spaces, such as in the example

of Caoyang Park, where the activity spaces for children and elderly groups are rigidly segregated. ② The population structure of Caoyang New Village is imbalanced: young and middle-aged people account for the largest number, elderly come second, and children last. The population ratio is close to 6:4:1.

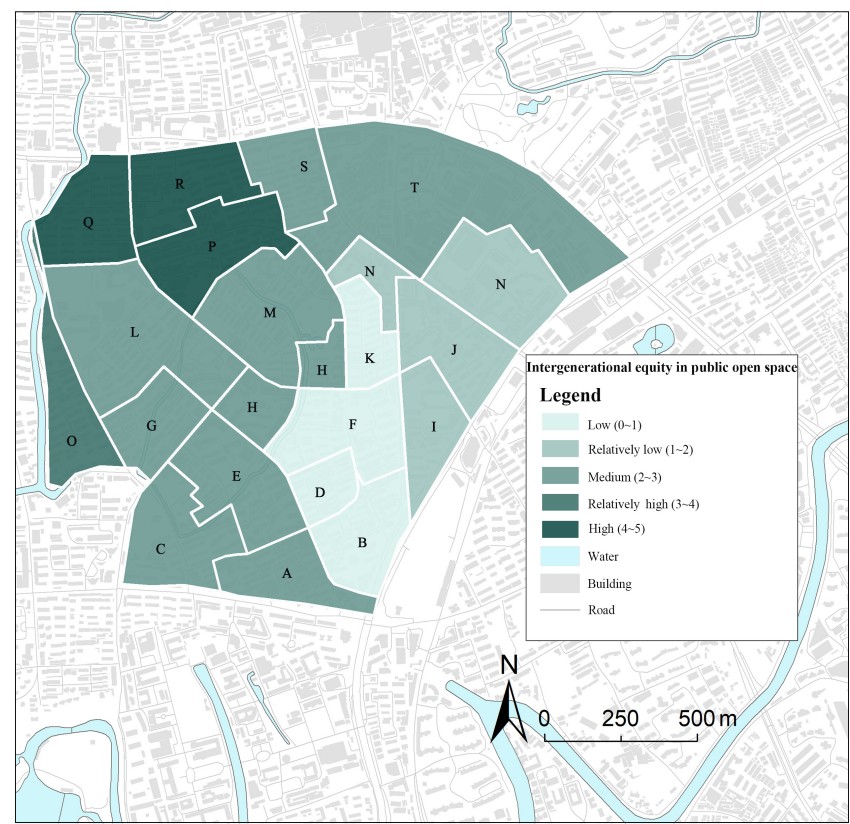

**Figure 13.** Intergenerational equity in POS. (Letters indicate different units of Caoyang New Village).

*4.3. Adaptation Space Share Is Much Higher for Children Than for the Young and Middle-Aged and the Elderly Groups*

The adaptation space share for each age group in the unit shows that children > the elderly > young and middle-aged people (Figure 14), with significant gap between the three groups. The adaptation space share for children in individual units is significantly higher than the average level, reaching a maximum of 11.34 m²/person, which is related to the continuous emphasis on child-friendliness in POS construction in recent years. However, the demand of children groups is much lower than that of the two other age groups, indicating that there is an obvious contradiction in "low demand–high adaptation space". The adaptation space share for the elderly and the young and middle-aged is generally not high, and the highest area of adaptation space share in a unit is only 2.77 m²/person for the elderly and only 1.93 m²/person for the young and middle-aged group, showing the opposing dilemma of "high demand–low adaptation space". Although the adaptation space share for the elderly has received more attention at present, the data show that the improvement of spatial adaptation for the elderly is not prominent. The young and middle-aged group, as the group with the greatest demand, has the lowest adaptation space share. The reasons could be that, on the one hand, the needs of young and middle-aged people are largely ignored in existing POS planning decisions, which overemphasize child-friendliness and adaptation for the elderly; on the other hand, the young and middle-aged people, as the mainstay of the family, spend less time using the space alone and usually appear as the guardians of children or old people. Because their activity space is mostly attached to the children and elderly, they have the lowest adaptation space share.

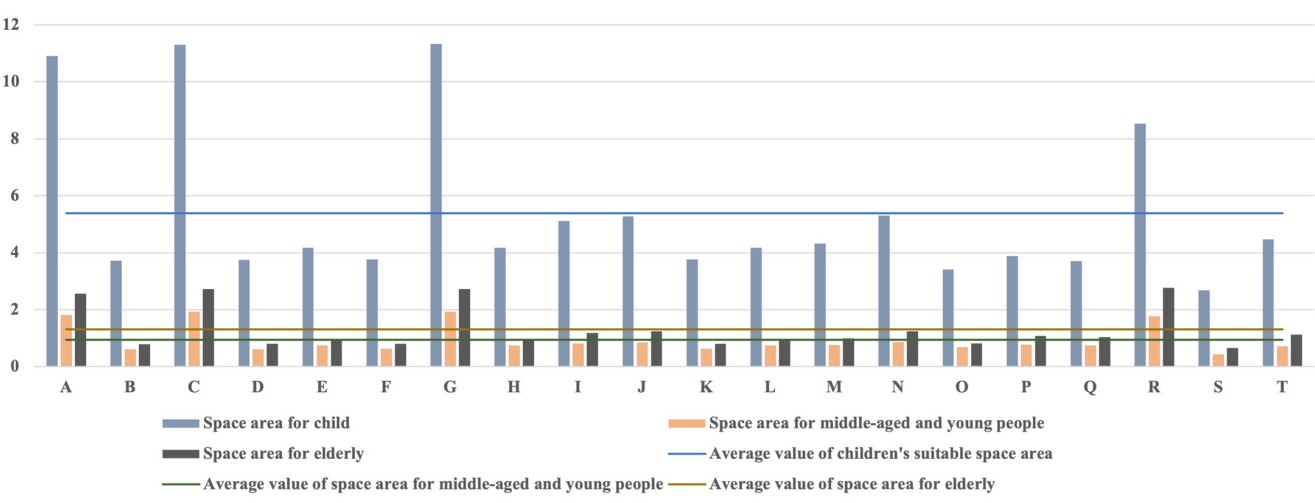

**Figure 14.** Adaptation space share for different age groups in the district of the residence committee. (Letters indicate different units of Caoyang New Village).

*4.4. Spatial Alienation between Comprehensive POS Supply Level and Intergenerational Equity Are Obvious*

The results (Figure 15) show that 65% of the units had a high comprehensive POS supply level but less intergenerational equity, with 6 units having an even less-than-average intergenerational equity level; 35% of the units had high-level intergenerational equity but a low comprehensive POS supply level, with only 1 unit having a comprehensive POS supply level above the average. The results indicate the following: First, there is no positive relationship between comprehensive POS supply level and intergenerational equity in Caoyang New Village; a high level of comprehensive POS service level does not mean great intergenerational equity. Second, the decision makers for allocation planning of the community POS here are paying inadequate attention to the differences in the population age structure among the units. In future planning decisions, on the basis of ensuring a comprehensive supply of POS, the age proportion of the population in the community's 15-min living circle should be fully considered, and activity spaces and facilities should be configured differently to achieve all-age sharing.

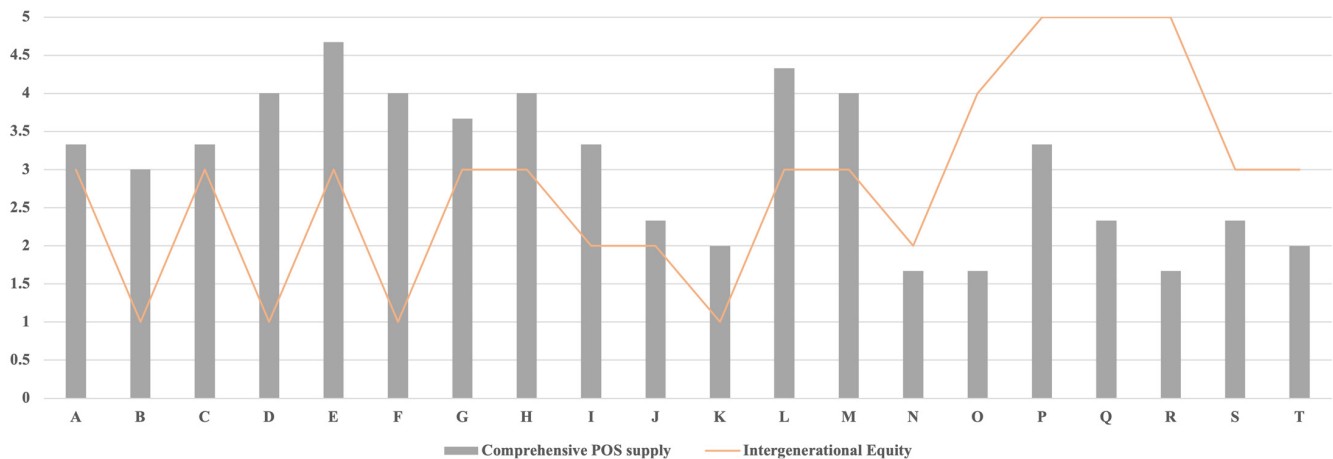

**Figure 15.** Comprehensive POS supply and intergenerational equity. (Letters indicate different units of Caoyang New Village).

**5. Discussion**

In the context of building an all-age friendly community, our research explored the intergenerational equity of all-age POS. We selected the 15-min living circle as the scope

of our research because administrative boundaries are rarely ever representative of the scope of residents' everyday lives. In addition, we comprehensively assessed the supply of POS in three dimensions—quantity, spatial distribution, and quality—allowing a relatively comprehensive evaluation of the POS supply. In particular, our research introduced the "adaptation space share" to measure the supply of adaptation POS for different age groups, which represents an improvement of the equity evaluation system to achieve the transformation of POS evaluation from "space supply" to "space adaptation supply". We used such methods as POI big data, Internet map APIs, and on-site surveys, ensuring a large degree of accuracy in the measurement results. Furthermore, in order to meet realistic development needs, we divided the POS into different types in terms of activity supply, environmental quality, convenience facilities, and space security.

As a result of the study, we found that the comprehensive spatial supply of Caoyang New Village is relatively high, but the intergenerational equity level is medium. Moreover, there is no positive relationship between the comprehensive supply of POS and intergenerational equity. This result reflects that in the explosive spatial growth and pattern, full consideration of spatial equity is lost; instead, the efficiency of spatial resource use is reduced, which induces a series of social problems [93]. In the future, first, we can make full use of three types of POS to balance the areas with insufficient space supply, for example, opening up some of the enclosed affiliated green spaces, such as the affiliated green space in front of Caoyang Second Middle School. Meanwhile, focus on intergenerational integration of space activities, such as children and elderly; these two groups of people present spatial complementarity and temporal compositeness in spatial use [94], so activity spaces could be upgraded by combining conversation-type space for the elderly with children's activity-type space and using landscape facilities, such as corridors, as a spatial transition to achieve intergenerational sharing (Figure 16).

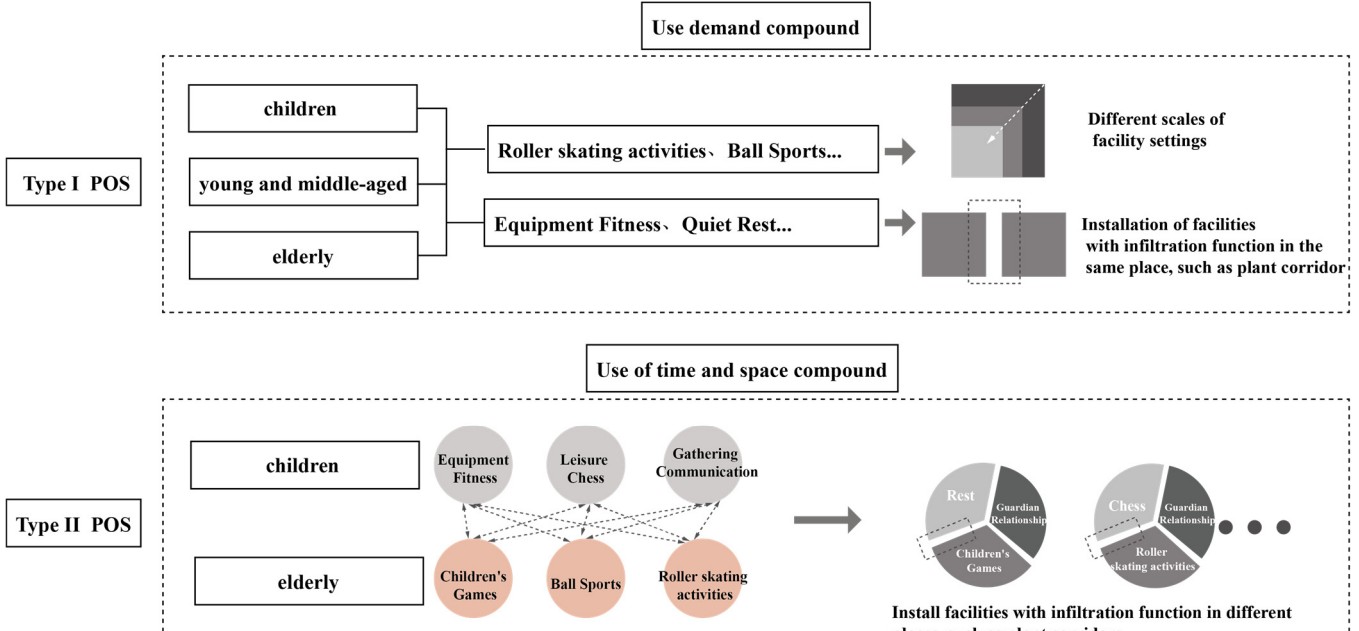

**Figure 16.** Optimization strategies for type I and type II spaces based on intergenerational equity enhancement.

By comparing the adaptation space share for different age groups, we found that the space renewal over-emphasized the child and elderly friendliness and ignored the needs of the young and middle-aged groups. This result reveals, to some degree, the problems in the current renewal. Yet, there are several issues here that need further discussion. First, does an equal area of adaptation space share for different age groups really mean equity? We need to further explore the differences in the intensity of space needs of different age

groups; for example, young and middle-aged groups do not spend much time using space, thus the configuration of their adapted space can be slightly lower than that of the other two groups, focusing more on the convenience of using space. Second, whether or not the space facilities are really suitable for different age groups, we need further research on facility usage satisfaction based on the quantification of physical space. Here, due to COVID-19, POSs were controlled and there were almost no active people in the POS, so it was difficult for us to conduct satisfaction research on the use of 28 POS's facilities, so only a sample of different types of POS's facilities satisfaction were studied. The activity preferences and spatial choices of people in different age groups are basically in accordance with those shown in Table 2. Yet, residents are more satisfied with the use of facilities in Type I spaces, followed by Type II and Type III spaces. This means that spaces of a slightly smaller scale in the renewal need to focus even more on the improved configuration of facilities.

Moreover, in renewal, the quality of POS is usually ignored, and the focus is simply on completing government instructions. From the perspective of all-age sharing, the POS planner should shift their role from a traditional resource allocator to the coordinator [95]. Planners should reasonably organize different types of POSs; for example, Type I and Type II spaces are used as the main activity spaces, while Type III spaces are used as supplementary activity spaces according to the differences in the needs of different age groups (Figure 17). Under the context of stock optimization, community renewal is seen as a process of community space reproduction. Daily life practices could be used as resources to organize the POS with the concept of "sharing"; clarifying the relationship between rights, responsibilities, and benefits of the shared space, a continuous community regeneration path is practiced in which multiple participants autonomously share governance [96].

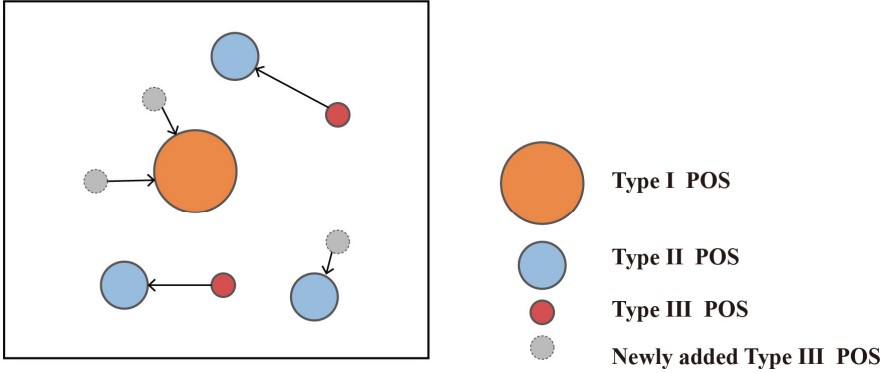

**Based on the difference in demand intensity of different age groups, the type III of space are used as a supplement to the activity space of the type I and type II of space**

**Figure 17.** Optimization strategies for type III.

## 6. Conclusions

At present, the POS renewal of the Caoyang New Village community is considered an exemplar in old communities in high-density cities in China. However, it still suffers from uneven and inequitable allocation of POS and is also representative of the problems that exist in the general community POS in China. This means over-reliance on traditional renewal theory may mislead practice due to demographic imbalance [4]. Through our research, we have found that some urban renewal models distribute all spatial attributes evenly across different areas. In the current renewal of old communities in high-density cities, the social equity level has been ignored. Despite the good intentions of the bottom-up planning approach, practical implementation in developing countries faces many problems, including opposition from local leaders, lack of awareness by residents, and lack of effective communication between government and residents [97], as well as the inaccessibility of complex land use planning models to local stakeholders, lack of communication platforms for negotiations between multiple stakeholders [98], and vague methodological guidelines

and limited implementation capacity [99]. In particular, facing the contradiction between the motivation of the government and the residents, the government's opinion usually dominates the decision-making process, where resident participation is merely symbolic. There is still a long road ahead for POS to be transformed in such a way that it can provide basic amenities equally for all citizens. Faced with the requirements of POS humanization, it is far from enough to emphasize only age-friendly design and child-friendly space. Under the requirements of high-quality urban development, the "public attributes" of POS need to be further explored, and people from different identities and ages should enjoy the same access to use space. In the process of re-engineering POS, people's needs and well-being in planning, design, construction, and governance need to be fully considered.

We have evaluated the comprehensive supply of POS more comprehensively in terms of quantity, layout and quality. However, the factors influencing the supply of POS go far beyond these three indicators, and in the future, more factors influencing the supply, such as transportation modes and regional economic differences, need to be taken into account. We introduce the concept of "adaptation space share" to evaluate the equity among different age groups and use the coefficient of variation to present the result of equity. Yet, the study only considered the suitability of the space facilities. In future study, indicators such as the spatial opportunity index and satisfaction from different age groups can be added to improve the system for evaluation of intergenerational equity. In addition, the measurement of POS intergenerational equity in high-density-city old communities in Shanghai and even in the Yangtze River Delta, can be increased to establish a universal intergenerational equity evaluation system to promote all-age sharing of POS.

**Author Contributions:** Conceptualization, Z.Z. and X.T.; methodology, Z.Z.; software, Z.Z.; validation, Z.Z.; formal analysis, Z.Z.; investigation, Z.Z.; resources, Z.Z.; data curation, Z.Z.; writing—original draft preparation, Z.Z.; writing—review and editing, X.T. and Y.W.; visualization, Z.Z.; supervision, Z.Z.; funding acquisition, X.T. All authors have read and agreed to the published version of the manuscript.

**Funding:** This research was funded by the National Social Science Fund of China, grant number 20BGL215.

**Data Availability Statement:** Not applicable.

**Conflicts of Interest:** The authors declare no conflict of interest.

## Notes

[1] The POST instrument and manual is available at https://www.science.uwa.edu.au/centres/cbeh/projects/post, accessed on 30 May 2023.

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
