# Peer review of "Evaluation of the Intergenerational Equity of Public Open Space in Old Communities: A Case Study of Caoyang New Village in Shanghai"

_land, doi:10.3390/land12071347_

Round 1

Reviewer 1 Report

The paper presents a complex and interesting subject of the social attractiveness of public open  spaces for different age groups, analyzing the aspect of quantity, quality and spatial distribution of POS in a densely built urban environment. However, there are several aspects that need to be clarified and developed in this paper.

The title

The title needs to be corrected and shortened. The repetition of similar information should be removed, the term "Evaluation of the Intergenerational Equity" contains the semiotic message of the term " all-age Sharing Perspective".  It is unreasonable to duplicate pretty much the same information using only different wording.

1. Introduction and 2. Literature Review

First of all the paper lacks a formulation of what is the scientific purpose of the study. The aim of the research is not presented in a proper way. The purpose and scope of the subject of the research should be defined clearly in this chapter.

The concept of "Intergenerational equity" as a main issue of the paper has not been comprehensively explained, and there is also insufficient bibliography, especially according to an urban perspective (only one item of literature - nr 24).

3. Methods

A Research Framework chapter (3.1)  requires a more detailed and specific discussion of the  three indicators  that are intended used to  comprehensively characterize the level of POS supply in the community - quantity,  spatial distribution, and quality. Furthermore the concepts of "open space opportunity index" , "service location entropy" and "POS quality service level" should be decently discussed and explained. 

The criteria for evaluating the quality of space (for POS) and the rationale for their selection need to be discussed. Also the concept of "adaptation space share" which was introduced to construct an intergenerational equity evaluation system require providing sufficient background and references.

In table 2  the Chinese characters have been left out, making the content difficult to understand.

4. Results

The results are accurately presented, but the numerous and very long names of the districts (Fig. 12, 13, 14) blure the reception of the diagrams and tables and makes it difficult to focus on the essence of the matter. I would suggest labeling the districts with symbols (e.g. A<B<C etc.) , which will facilitate orientation and understanding of the content. 

5. Optimization Strategies and 6. Discussion and Conclusion 

The chapter is quite limited in volume, contains some general suggestions for changes, e.g. regarding the rearrangement of the layout or the quality of POS  , and seems not to require a separate section, but should be integrated into the Discussion and Conclusion Chapter.

It would also be worthwhile for the authors to be tempted to make conclusions and suggestions not only about the place of research but also ones that could have a more general reference, which could increase the article's contribution to world science in the fields of urban planning as well as sociology.

Author Response

Dear Reviewer:

Thank you very much for your consideration and comments on our manuscript“Evaluation of the Intergenerational Equity of Public Open Space in Old Communities from an All-age Sharing Perspective: a Case Study of Caoyang New Village in Shanghai”. We are very grateful to receive your comments which are very helpful for improving our paper. We have sincerely and carefully considered the comments and revised the manuscript thoroughly. Detailed point-by-point response is given below. All revised portions have been marked up using the “Track Changes” in the revised manuscript.

I hope that the revised manuscript is suitable for publication in Land. Thank you again for your help with the manuscript. I am more than happy to provide any further information you may need.

 (The details are in the attachments)

Reviewer 2 Report

The topic of the dissertation is relatively well researched. The researcher has focused on some real world problems. I also recognize the workload of the researcher. To better the quality of the dissertation, I have some suggestions:

1) The article is a case study, does it need a fuller explanation in case selection. If it is only 1 community, then the title is suggested to be placed directly in the main title, not as a subtitle.

2) The 9 pages of serial numbers are a bit confusing, and the serial number hierarchy should be sorted out again.

3) The results in part 4 are a bit repetitive with the conclusions in 6, and the title is suggested to be adjusted.

(4) The reference basis of Table 2 is not very convincing, and it is suggested that this part should be explained again.

5) In part 5, the optimization strategy is too simple and no very effective strategy is proposed.

6) The research methodology needs to be explained more fully.

7) The reference format is standardized again.

Reading the whole paper, I think this paper can be published after the above problems are solved.

 Minor editing of English language required

Author Response

Dear Reviewer:

Thank you very much for your consideration and comments on our manuscript“Evaluation of the Intergenerational Equity of Public Open Space in Old Communities from an All-age Sharing Perspective: a Case Study of Caoyang New Village in Shanghai”. We are very grateful to receive your comments which are very helpful for improving our paper. We have sincerely and carefully considered the comments and revised the manuscript thoroughly. Detailed point-by-point response is given below. All revised portions have been marked up using the “Track Changes” in the revised manuscript.

I hope that the revised manuscript is suitable for publication in Land. Thank you again for your help with the manuscript. I am more than happy to provide any further information you may need.

 (The details are in the attachments).

Round 2

Reviewer 1 Report

The authors have sincerely and carefully considered the comments and revised the manuscript. All my comments and suggestions for correction and clarification of unclear issues and terms have been considered and implemented. The paper can be published as submitted.